# Murine and Human Gastric Tissue Establishes Organoids after 48 Hours of Cold Ischemia Time during Shipment

**DOI:** 10.3390/biomedicines11010151

**Published:** 2023-01-06

**Authors:** Daniel Skubleny, Saurabh Garg, Jim Wickware, Kieran Purich, Sunita Ghosh, Jennifer Spratlin, Dan E. Schiller, Gina R. Rayat

**Affiliations:** 1Department of Surgery, Faculty of Medicine and Dentistry, University of Alberta, Edmonton, AB T6G 2R3, Canada; 2Department of Oncology, Faculty of Medicine and Dentistry, University of Alberta, Edmonton, AB T6G 2R3, Canada; 3Department of Mathematical and Statistical Sciences, Faculty of Science, University of Alberta, Edmonton, AB T6G 2R3, Canada

**Keywords:** stem cell, viability, dose response, procurement, gastric cancer

## Abstract

An inadequate supply of fresh tissue is a major limitation of three-dimensional patient-derived gastric organoid research. We propose that tissue procurement for organoid culture could be increased by developing a cold storage shipment protocol for fresh surgical tissues. Sixty stomach specimens from C57BL/6J mice were resected, of which forty-five were stored in Hank’s Balanced Salt (HBSS), University of Wisconsin (UW), or Histidine-Tryptophan-Ketoglutarate (HTK) solutions for subsequent organoid culture. Stomachs were dissociated and processed into gastric organoids as fresh tissue or after transport at 4 °C for 24 or 48 h. All gastric organoid cultures were established and maintained for 10 passages. Cold storage for 24 or 48 h did not significantly affect organoid viability. Although cold storage was associated with decreased organoid growth rate, there were no differences in viability, cytotoxic dose response, or LGR5 and TROY stem cell gene expression compared to organoids prepared from fresh tissue. As a proof of concept, six human gastric cancer organoids were established after 24 or 48 h of storage. Patient-derived gastric organoids from mouse and human gastric tissue can be established after 48 h of cold ischemia. Our method, which only requires ice packs, standard shipping containers, and HBSS is feasible and reliable. This method does not affect the reliability of downstream dose–response assays and maintains organoid viability for at least 10 passages. The shipment of fresh tissue for organoid procurement could serve to enhance multicenter collaboration and achieve more elaborate or controlled organoid experimentation.

## 1. Introduction

Organoid culture is a promising tool to advance translational medicine. Patient-derived organoids have been demonstrated to recapitulate parent tumor histology and molecular characteristics with higher fidelity compared to traditional two-dimensional cell culture [1]. Organoids may be utilized longitudinally by creating biobanks and can be perpetuated in vitro or in vivo as xenografts in immunodeficient mice [2]. In oncology, in vitro patient-derived organoid dose response to anti-cancer therapy has been shown to correlate with clinical recurrence-free survival [3].

Patient-derived organoid culture is a specialized and resource-intensive technique [4]. Organoid development requires the procurement of fresh tissue via biopsy, needle biopsy, or surgical specimens [5,6]. Even with modern protocols, the successful establishment of tumor organoid culture ranges from 16% to 100% depending on tissue origin [7,8]. In contrast to biologically homogenous immortal cell lines, organoid research requires larger sample sizes to demonstrate a meaningful biological signal due to increased heterogeneity from patient-derived samples. Thus, tissue resources for organoid culture must be optimized for efficient and reliable testing to advance our understanding of organoid models and their applicability to the clinical realm.

We propose that tissue procurement for organoid culture could be increased by developing a cold storage shipment protocol for fresh surgical tissues or biopsies. In this scenario, increased organoid sample sizes could be achieved, and more elaborate or controlled experimentation could occur if tissue for organoid growth is routed to fewer specialized laboratories. In our study, we investigated the feasibility of shipping fresh gastric tissue on cold packs over 24 or 48 h for the purpose of establishing organoid culture. We assessed the effects of various shipping media (Hank’s Balanced Salt Solution (HBSS), University of Wisconsin solution (UW), and Histidine-Tryptophan-Ketoglutarate solution (HTK)) and dissociation times (Fresh, 24 h, and 48 h) on organoid viability, growth rate, LGR5^+^ and TROY^+^ stem cell populations, and anti-cancer drug response.

## 2. Materials and Methods

### 2.1. Study Design

The study design is illustrated in Figure 1. Briefly, we collected sixty gastrectomy specimens from C57BL/6J mice divided across 5 experimental cohorts. Twelve mice specimens were included in each cohort. Three mouse stomachs used for in vivo controls were divided longitudinally and then immediately processed for molecular and histology analyses. The nine remaining stomachs were cleaned in sterile PBS, and one each was placed in HBSS, UW, or HTK and dissociated immediately as fresh tissue or mock-shipped on ice packs in Styrofoam coolers and dissociated after 24 or 48 h. Of note, the first cohort included 6 mice, and stomachs were divided in half. Due to an inadequate number of dissociated cells, all cohorts thereafter used whole stomachs for each experimental sample. The compositions of the HBSS (Sigma-Aldrich), UW (Belzer), and HTK (Global Transplant Solutions) are provided in the Appendix A.

Organoids were maintained for a total of 10 passages. Specimens for gene expression and histology were collected from control (i.e., non-dissociated); initial dissociation; and passage 1 and 5 samples. A dose–response drug assay was conducted on all nine gastric organoids in cohort 4 from passages 2 and 6. In addition, 11 human gastric cancer biopsies were mock-shipped for 24 and 48 h, respectively, and were processed for human organoid culture.

### 2.2. Mouse and Human Specimens

C57BL/6J mice (Jackson Laboratory, Bar Harbor, Maine, US) were retrieved from breeding colonies maintained by Health Sciences Laboratory Animal Services at the University of Alberta. Prior to gastrectomy, these mice were cared for according to the guidelines established by the Canadian Council on Animal Care. Ethics approval was granted by the University of Alberta Research Ethics Office under a Category A Exemption based on the principle of reduction.

All human clinical participants consented according to the approved ethics protocol granted by the Health Research Ethics Board of Alberta (study ID: HREBA.CC-17-0228). Specimens were obtained via endoscopic biopsy taken at the time of screening laparoscopy prior to any chemotherapy intervention.

### 2.3. Organoid Culture 

Our organoid protocol closely approximated previously characterized intestinal organoid protocols from Bartfeld et al. and Mowat et al. [9,10]. Specific reagent information and catalogue numbers can be found in Appendix A. Briefly, stomachs were resected from mice; washed in sterile PBS; and placed in 50 mL conical tubes containing HBSS with 200 units/mL penicillin and 200 µg/mL streptomycin, UW, or HTK. Tissues were minced with sterile scissors and washed three times in sterile PBS. Minced tissues were enzymatically and mechanically digested in 20 mL digestion buffer (Advanced DMEM/F12, 100 units/mL penicillin and 100 µg/mL streptomycin, 2.5 µg/mL amphotericin B, 2.5% FBS, 75 units/mL Collagenase XI, and 125 µg/mL Dispase II) and placed in a mechanical water bath at 37 ^o^C for 1 h. Next, tissues were pipetted aggressively up and down 10 times with a 10 mL pipette to dissociate cells and filtered through a 70 µm strainer coated in 10% FBS/PBS. Cells were washed and suspended in DMEM. A 15 µL aliquot was mixed 1:1 with Trypan Blue (Gibco, 15250061) and counted on a hemocytometer. Cells were resuspended in ice-cold 70% Matrigel (Corning, 356253) in Advanced DMEM/F12 at a concentration of 1000 cells/µL, and 35 µL Matrigel domes were placed in a prewarmed 24-well tissue culture treated plate. Organoids were cultured in 500 µL organoid culture medium at 37 ^o^C and 5% CO_2_ until mature for splitting or downstream analysis. Organoid growth and media were assessed daily, with media changes occurring every other day unless significant cellular debris or colorimetric change was present.

Organoid culture media contained 1:1 basal culture medium and conditioned L-WRN cell supernatant enriched in R-spondin, noggin, and Wnt (ATCC, CRL-3276). Conditioned L-cell supernatant was prepared according to Miyoshi and Stappenbeck [11]. Organoid culture media contained Advanced DMEM/F12, 2 mM L-Glut, 10 mM HEPES, 100 units/mL penicillin and 100 µg/mL streptomycin, 2.5 µg/mL amphotericin B, 1X N2, 1X B27, 1 mM N-acetylcysteine, 1 nM gastrin, 10 mM nicotinamide, 500 nM A83-01, 10 µM SB202190, and 50 ng/mL mouse epidermal growth factor (EGF). Human organoid culture used human EGF and, for the first plating only, 10 µM Y-27632.

Mature organoids were passaged approximately every 5 days depending on the abundance of cystic organoids and cell sloughing. Briefly, ice-cold Advanced DMEM/F12 was added to Matrigel domes and mechanically lifted using a pipette tip. Organoids were dissociated using TrypLE Express (Gibco, 12604013), washed three times, and counted with a hemocytometer. Organoid cells were then cultured or allocated for downstream analysis. Organoid viability and relative growth rate were measured using a hemocytometer with Trypan Blue for each passage. The relative growth rate was calculated assuming exponential growth.

### 2.4. Immunofluorescence and Immunohistochemistry

Tissue specimens for hematoxylin and eosin (H + E) staining and immunofluorescence (IFC) of whole mouse stomach or organoid-containing Matrigel domes were fixed in zinc-formalin (Zinc Formal-Fixx, Thermo Scientific) for 24 h and stored in 70% ethanol prior to preservation in paraffin. Fixed organoids were suspended in agar prior to paraffin embedding. Specific antibody combinations, dilutions, incubation times, and antigen retrieval buffers are listed in Appendix A. Briefly, 5 µm tissue sections were deparaffinized in Histoclear (National Diagnostics) and rehydrated. Microwave heat-induced epitope retrieval was performed using Sodium Citrate (pH 6, heated to 94 °C in 1 min intervals followed by 9 min continuous heat). Permeabilization was performed with 0.5% Triton X-100 in PBS. Non-specific epitopes were blocked using 10% normal goat serum in PBS. Tissue sections were stained with primary antibodies (rabbit anti-pan cytokeratin/mouse anti-MUC5AC, rabbit anti-pan cytokeratin/mouse anti-TROY, and rabbit anti-pan cytokeratin/mouse anti-LGR5) and secondary antibodies (anti-rabbit IgG Alexa Fluor 488 (green) and anti-mouse IgG Alexa Fluor 568 (red)). Autofluorescence was diminished using a TrueView Quenching kit per the manufacturer’s protocol (Vector, SP-8400-15). Nuclear counterstaining was performed with DAPI (blue) followed by cover slipping with Vectashield Vibrance Antifade mounting media (Vector, H-1700). Images were captured on an AxioCam HRc camera and processed using ImageJ [12].

Immunohistochemistry was performed on human gastric organoids. Tissues were rehydrated, and antigen retrieval was performed as above. Endogenous peroxidases were blocked with 3% hydrogen peroxide in methanol. Non-specific binding was mediated by blocking with 20% normal goat serum in PBS and avidin/biotin blocker per the manufacturer’s protocol (Vector Laboratories-2001). MUC5AC primary antibody was diluted 1:250 and incubated at 4 °C overnight. Biotinylated IgG secondary antibody was incubated at 1:200 dilution for 30 min at room temperature. Antibody detection was performed using avidin-biotin complex/horseradish peroxidase (Vector Laboratories) and 3,3-diaminobenzidine tetrahydrochloride (DAB, Abcam, ab64238) per the manufacturer’s protocol. Tissues were counterstained with hematoxylin. Images were captured using a Leica Aperio CS2 digital slide scanner and selected using QuPath version 0.3.1.

### 2.5. qRT-PCR

RNA was extracted from whole mouse stomach and organoids using TRIzol Reagent according to the manufacturer’s protocol (Invitrogen, 15596026). Whole mouse stomach was disrupted using a bead mill homogenizer. RNA concentration and purity were assessed using a NanoDrop 1000 (Thermo Scientific). Only samples with an A260/280 greater than 1.8 were included for analysis. Isolated RNA was stored at minus 80 °C with 0.2 units/µL SUPERASE-In RNase inhibitor (Invitrogen, AM2694). mRNA was reverse-transcribed into cDNA using 100 ng mRNA with a High-Capacity RNA-to-cDNA Kit (Applied Biosystems, 4387406). Quantitative real-time PCR was performed in a 96-well CFX Connect Bio-Rad RT-PCR Detection System using 10ng input cDNA; TaqMan Fast Advanced Master Mix (Applied Biosystems, 4444964); and stock TaqMan PCR primers from ThermoFisher Scientific (Tnfrsf19/TROY (Mm00443506_m1), LGR5 (Mm00438890_m1), and beta-actin/ACTB (Mm01205647_g1)).

PCR amplification efficiency was assessed to establish optimal input cDNA concentration and to validate the assay for our selected transcripts (Appendix A). We conducted a validation experiment to confirm the equal efficiency of the target and housekeeping genes (Appendix A) [13]. Relative gene expression levels were calculated using the comparative CT method (delta delta/ΔΔCT) using pooled passage 8 mouse organoids as reference. Gene expression values were subsequently scaled to the mean of whole stomach controls prior to downstream analysis.

### 2.6. In Vitro Dose–Response Assay

Our combination FOLFOX (5-fluorouracil, oxaliplatin, and leucovorin) dose–response assay was validated using the human gastric cancer cell line AGS (ATCC CRL-1739). AGS cells were cultured in Ham’s F-12K (Gibco), 10% FBS (Gibco), 100 units/mL penicillin, and 100 µg/mL streptomycin (Gibco). Briefly, 5000 cells were plated in 96-well plates and grown for 24 h. Anti-cancer drugs 5-fluorouracil (Tocris, 3257) and Oxaliplatin (Tocris, 2623) were added in triplicate over 8 half-log dilutions with 800 µM and 2400 µM initial concentrations, respectively. A single 500 µM dose of Leucovorin (Toronto Research, L330400) was added to each treatment well. Cells were treated for 48 h, followed by a CCK-8 viability assay (Abcam, ab228554), which was performed according to the manufacturer’s protocol. We replicated our assay in three independent trials to assess reproducibility.

For organoid dose–response assays, organoids were passaged and dissociated according to our protocol. As above, 5000 cells were plated in 96-well plates and grown in organoid media for 24 h, followed by 48 h of FOLFOX treatment and viability assessment using a CCK-8 assay.

### 2.7. Statistical Analysis

Statistical analyses were completed using R version 4.1.2 and Prism 9.3.1 [14]. Differences between three or more groups were assessed using Kruskal–Wallis test, with Dunn’s post hoc test for multiple comparisons when applicable. Multiple comparison corrections were performed using the Bonferroni method. A paired Wilcoxon test was used to assess paired dose–response data. We used Pearson’s correlation to assess the relationship between gene expression and dose–response data. Categorical data were assessed with Fisher’s exact test. A one-way ANOVA was used to compare viability and growth rate. Multivariable generalized additive models (GAMs) of growth rate, viability, and stem cell gene expression were created to determine the independent effects of media, cold storage/dissociation time, cohort, and passage number. A series of models were analyzed and compared using likelihood ratio tests of nested models. We used the variable inflation factor to assess collinearity. GAM fits were tested relative to multiple linear regression and linear regression with relevant spline terms for continuous variables. Predictions from these models were used to create adjusted response plots via ggplot2.

Dose–response data were processed and analyzed in GraphPad. First, media control absorbance values were subtracted from all experimental wells. Absorbance values were then normalized between 0 and 100%. Mean drug concentrations from 5-fluorouracil, oxaliplatin, and leucovorin were log10-transformed. Next, outliers were removed using Q = 1%, least-squares variable slope non-linear regression estimated dose–response curves, and half-maximal inhibitory concentration (IC50).

## 3. Results

### 3.1. Gastric Organoids Recapitulate Murine Gastric Tissue and Morphology

We successfully established gastric organoids from 45 mice (25 male, 20 female) with a median age of 129 days (interquartile range (IQR) 119,140). The median number of days between passages was five (IQR 3,6), and the median total time spent in culture was 54 (41,56) days. We assessed the morphology of gastric organoids from each set of medium and dissociation time conditions relative to the whole mouse stomach using hematoxylin and eosin staining. Figure 2A shows representative images that demonstrate the recapitulation of tissue morphology. Irrespective of medium or dissociation time, organoids from passages 1 and 6 formed circular hollow structures with cells bound by a basement membrane reminiscent of gastric epithelial tissue. We confirmed that our organoids were of gastric origin by assessing the presence of MUC5AC, which is exclusively present in normal gastric tissue (Figure 2B) [15].

### 3.2. Dissociation Viability Is Affected by Storage Medium and Is Cohort-Dependent

The initial cell viability following dissociation was assessed using Trypan Blue dye and a hemocytometer. The univariable analysis of dissociation viability for all organoids is displayed in Figure 2C–E with respect to medium, cold storage time, and each dissociation procedure/cohort. Cell viability was significantly decreased in tissues shipped in HTK relative to HBSS (Dunn’s test, HBSS-HTK adjusted *p* = 0.01). Of note, the isolated analysis of fresh tissue dissociation viability found no significant differences between storage media (Appendix A). Furthermore, for each storage medium, there were no significant differences in viability between dissociation times (Appendix A). Significant variability in dissociation viability was identified between cohorts (Kruskal–Wallis, *p* < 0.01).

We constructed a multivariable linear regression model to evaluate the effect of measured confounders and our experimental groups on dissociation viability following cold storage for 24 and 48 h (Table 1 and Appendix A). Of note, no difference in cell viability was found between cells dissociated after 24 or 48 h of cold storage compared to fresh tissue. HTK and cohorts 4 and 5 were significantly associated with decreased cell viability relative to HBSS and cohort 1, respectively (*p* < 0.05).

### 3.3. Long-Term Organoid Viability Is not Affected by Storage Medium and Dissociation Time

Organoid cell viability was measured during each passage to assess the overall health of organoids following cold storage. Of note, we established viable mouse organoid cultures in 100% of the experimental conditions, and all organoids were propagated for a total of 10 passages. Univariable analysis found no significant difference in pooled viability with respect to storage medium and dissociation time (*p* < 0.05) (Figure 3A left and right, respectively).

Next, we assessed the cell viability over the course of 10 organoid passages. Given the significant cohort-dependent effects related to dissociation viability, we examined the trend of viability for each cohort (Appendix A). We observed heterogenous nonlinear variation in viability between cohorts; however, there were no obvious effects related to storage medium or dissociation time (Appendix A).

Using likelihood ratio tests, we developed a generalized additive model (GAM) to best approximate the relationship between viability, our experimental conditions, and relevant confounding variables (Figure 3D). Penalized cubic regression splines were fit using generalized cross validation for passage number, days in culture, and relative growth rate. We visually present the adjusted effects of storage medium and dissociation time on viability in Figure 3C. Of note, despite differences in storage medium and dissociation time, organoid viability fell within a fairly narrow range between ~65 and 85%, with small 95% confidence intervals for the loess smoothing functions. We found no significant difference in viability between UW or HTK and HBSS (*p* < 0.05). We identified a small, yet statistically significant increase in cell viability for organoids dissociated after 24 and 48 h of cold storage relative to fresh organoids (*p* < 0.001). The bottom panel of Figure 3C demonstrates that 24 h and 48 h organoids had a relatively linear relationship with viability over successive passages, whereas fresh organoids exhibited a sinusoidal pattern.

Once again, we found that specific cohorts exerted significant batch effects on cell viability (Figure 3D). The graphical representation of nonlinear covariates presented in Appendix A provides insight into the relationship between the significant variables of passage number, days in culture, growth rate, and cell viability. For example, passage number exerted a complex nonlinear effect on viability, whereas days in culture and growth rate were associated with relatively linear effects on viability.

### 3.4. Cold Storage for 24 or 48 Hours Decreases Organoid Growth Rate

The univariate analysis of the storage medium found no significant effect on pooled growth rate (*p* < 0.05) (Figure 3B, left). However, organoids derived from fresh tissues were associated with a significantly greater relative growth rate (Figure 3B, right) compared to organoids generated after 24 and 48 h of cold storage (Dunn’s test, fresh 24 hours adjusted *p* < 0.001, fresh 48 hours adjusted *p* < 0.001).

Compared to viability, growth rates over successive passages displayed greater similarity between cohorts (Appendix A). We fit a GAM to model the growth rate in keeping with the procedure outlined in the previous section. After adjusting for confounders, dissociation after 24 or 48 h remained significantly related to the decreased growth rate (Figure 3F and Appendix A). The bottom panel of Figure 3E displays this consistent effect across all 10 passages. Once again, the storage medium did not significantly affect the growth rate (Figure 3E (top) and 3F).

Variability in growth rate was significantly related to cohorts 4 and 5, viability, passage number, and the number of days in culture. In our model, viability was best interpreted as having a linear relationship with growth rate, whereby increased viability was associated with an increased growth rate. The complex nonlinear relationships for passage number and the number of days in culture are displayed in Appendix A.

### 3.5. In Vitro Gene Expression of Gastric Stem Cell Markers LGR5 and TROY Re-Establishes Baseline Endogenous Levels after 6 Passages and Is Unaffected by Cold Storage Conditions

We assessed the gene expression of essential gastric stem cell molecules LGR5 and TROY in whole stomach tissue, upon dissociation (passage 0) and after passages 1 and 6. No significant relationship was identified between LGR5 and TROY gene expression and organoid cold storage/dissociation time or storage medium (Figure 4A,B). As shown in Figure 4C,E, we identified significant differences in LGR5 and TROY gene expression between whole stomach tissue, dissociated tissues, and subsequent organoid passages. We examined the significance of the differences between each set of conditions with Dunn’s test. In both cases, LGR5 and TROY gene expression significantly decreased upon the dissociation of tissues relative to endogenous whole stomach levels. Stem cell gene expression significantly increased in successive passages and was statistically similar to endogenous levels following passages 1 and 6 (Figure 4C,E).

To confirm the translation of LGR5 and TROY, we assessed protein expression using immunofluorescence. In Figure 4D, the LGR5 expression is indicated in representative images of whole stomach tissue (left) and gastric organoids (right). LGR5 was predominately co-localized with pan cytokeratin expressed on cell membranes, which was consistent with prior studies [16,17]. TROY expression was also identified in whole stomach tissue and gastric organoids (Figure 4F, left and right, respectively). In both tissues, TROY was expressed mainly in the cytoplasm and cell membrane, although some expression was present in the nuclei [17].

Multivariable GAMs were established to assess TROY and LGR5 gene expression using mock-shipment covariates, cohort, and passage number. We modeled passage number as a continuous variable to evaluate the nature of the relationship between organoid culture and gene expression. As shown in Figure 5A, we observed a linear increase in TROY and LGR5 expression among dissociated tissues and organoids after passage 1. However, the rate of increase in stem cell gene expression subsided and appeared to reach a steady state by passage 4. Overall, the passage number contributed significantly (*p* < 0.001) to stem cell gene expression, and the relationship between LGR5 and TROY was nearly identical. Again, significant cohort batch effects were identified for LGR5 and TROY expression (Figure 5B,C). Storage conditions related to medium and dissociation time remained insignificant.

### 3.6. In Vitro Organoid Dose Response to Cytotoxic Therapy Is Significantly Associated with Passage Number and Stem Cell Gene Expression

Ideally, the shipment of tissues should increase organoid procurement and not affect downstream analyses such as dose–response assays. We performed a dose–response assay using FOLFOX cytotoxic therapy to assess possible effects related to cold storage. Using the AGS cell line, we validated the reproducibility of our drug assay. Across three independent drug assays, our dose–response curves were statistically similar (Appendix A).

The dose–response curves for a single cohort of organoids at passages 1 and 6 are shown in Figure 6A. We achieved reliable nonlinear estimates of all drug assays with an adjusted goodness of fit (R^2^) ranging from 0.82 to 0.99. A greater efficacy of cytotoxic therapy was observed in paired organoid lineages from passage 1 relative to passage 6 (Figure 6B, paired Wilcoxon, *p* < 0.01). No significant difference in dose–response IC50 was identified between dissociation times or storage media variables (Figure 6C,D). Within the same organoid lineages, dose–response IC50 values correlated strongly with LGR5 and TROY gene expression.

Next, we assessed the Pearson’s correlation of dose–response IC50 values with LGR5 and TROY gene expression measured from the same organoid lineages. Both LGR5 and TROY gene expression was strongly correlated with IC50 (Pearson’s R = 0.7, *p* < 0.01). In Figure 6E,F, it is shown that this association also corresponded to clear differences in expression between organoid passages.

### 3.7. Human Tumor Organoids can Be Established following 24- and 48-Hour Cold Storage in HBSS Solution

Endoscopic biopsies were retrieved from 23 gastric cancer patients (28 samples) and mock-shipped in cold HBSS to represent fresh, 24 h, and 48 h dissociation, respectively. All patients provided informed consent according to our ethics protocol approved by the Health Research Ethics Board of Alberta. An overview of the human gastric cancer organoids, dissociation viability, and culture status is presented in Appendix A. No difference in dissociation viability was observed between fresh dissociated human cancer organoids and those that were placed in cold storage for 24 or 48 h (Wilcoxon, *p* = 0.13) (Figure 6G). A total of 20 human gastric cancer organoids were established (success rate 71.4%), and no difference was observed in the successful establishment of fresh tissues versus those with at least 24 h of cold storage (success rate 82.4% vs. 54.5%, Fisher’s exact test, *p* = 0.2) (Figure 6H). Organoid morphology was consistent between fresh and cold-storage organoids in vitro (Figure 6I). Tumor-derived organoids recapitulated the morphology of parent tumor tissue in hematoxylin- and eosin-stained sections (Figure 6J). Furthermore, MUC5AC was expressed in organoid tissues, which confirmed the growth of gastric epithelium. Of note, organoids from 24 h shipped tissues recapitulated diffuse-type histology, with signet ring cells noted in the parent tumor.

## 4. Discussion

Using a mouse gastric organoid model, we demonstrated that gastric organoids can be established from tissue biopsies that have undergone cold storage for 24 or 48 h. Although HTK was associated with decreased viability following dissociation, no significant difference in viability or growth rate was found across 10 passages between HBSS, UW, and HTK. A decreased growth rate was identified in organoids established after 24 and 48 h of cold storage; however, this did not affect cell viability, LGR5 and TROY stem cell gene expression, or dose–response assays. These data support the notion that the cold shipment of fresh tissue on ice packs is a feasible approach to improve organoid tissue procurement.

Gastric organoid models are thought to provide high-fidelity models that will likely form the cornerstone of future translational oncology research [2,18]. Several pre-clinical organoid studies have identified a correlation between in vitro organoid dose-response and clinical tumor response [19,20,21,22]. Despite the optimism and promising preliminary findings, prospective evidence supporting the use of organoids as a translational model for personalized medicine is lacking. The first clinical trial assessing whether colon cancer organoids could be used to allocate clinical therapy failed to establish feasibility [6]. To improve organoid models, additional research is required. Thus, enhancing tissue procurement would eliminate a practical bottleneck that is preventing the expansion of organoid research.

Previous research has attempted to improve tissue procurement and preservation. Walsh et al. compared the effectiveness of flash freezing fresh tissue in liquid nitrogen to cryopreservation in dimethyl sulfoxide (DMSO) at −80 °C prior to organoid development [23]. In comparison to organoids derived from fresh unfrozen tissue, organoids developed from tissues cryopreserved in DMSO most accurately recapitulated dose response to various anticancer drugs. Another option to distribute organoid tissue involves shipping established organoid cultures from patient-derived tissue or induced pluripotent stem cells. Using live retinal organoid cultures, shipment protocols at room temperature or 37 °C have proven to maintain organoid viability [24,25]. We argue that neither of these methods is feasible for the widespread collection and distribution of organoid tissues. The first method requires liquid nitrogen, an adequate −80 °C freezer capacity, and DMSO within surgical facilities. Further, the shipment of these frozen tissues would require expensive and elaborate shipping conditions. The second method relies on establishing organoid culture prior to transport. This requires the availability of a nearby organoid research lab. Furthermore, any established cultures will be prone to greater batch variability between separate organoid labs. Our method only requires affordable HBSS, ice packs, and shipping coolers.

To our knowledge, we are the first group to characterize the gene expression of gastric stem cell markers LGR5 and TROY in gastric organoids starting from tissue dissociation through to subsequent passages. Given that organoid media purposefully drive stem cell growth, it is possible that organoids could become enriched in stem cells over successive passages. Our findings suggested that the drug-response assay results were associated with the passage number and/or stem cell gene expression. In either case, any effects on dose–response efficacy would need to be considered in translational organoid models to capture the true in vivo efficacy. Indeed, LGR5^+^ and TROY^+^ cells may give rise to cancer stem cells [26,27]. Increased LGR5 in human gastric cancer is associated with worse overall survival and a decreased tumor response grade following neoadjuvant chemotherapy [28]. The inhibition of LGR5 expression in the AGS cell line was also associated with enhanced in vitro dose-response. The future characterization of the relationship between increasing organoid passages, stem cell gene expression, and cytotoxic therapy efficacy is required.

Significant variability in experimental results between cohorts was another pertinent finding in our study. Despite using stomachs derived from inbred mice, we found cohort-dependent effects for dissociation viability, overall viability, and stem cell gene expression values. Our experimental method involving organoid dissociation, splitting, and growth was consistent across cohorts. For example, we used consistent volumes of Matrigel and cellular concentrations. We also used the same batch of L-WRN cell supernatant, which contained the stem cell growth factors R-spondin, noggin, and Wnt. This finding suggested that even subtle differences in the surgical handling of tissues and subsequent dissociation processes can lead to heterogeneity in a given organoid culture. Limited research has evaluated batch consistency within identical clinical/experimental samples. In their study, Little et al. identified significant batch variability between kidney organoids generated from the same induced pluripotent stem cells [29]. We were unable to determine whether the cohort-dependent effects were established upon initial dissociation and maintained throughout the organoids’ life span or whether multiple instances of heterogeneity accumulated over time (or both). Nonetheless, these findings suggest the need for additional standardization in the generation of gastric organoids and the importance of accounting for confounding factors related to batch variability.

In this study, we established that the shipment of fresh gastric tissue from mice and humans is a feasible and reliable method to increase the procurement of primary organoid tissue. Our shipment method uses ice packs, standard shipping coolers, and HBSS supplemented with penicillin and streptomycin. Our method is relatively simple to implement in a clinical setting because it does not require liquid nitrogen or minus 80 °C fridges. In addition to the methodological feasibility, we showed that the shipment conditions did not affect the reliability of downstream dose–response assays and that the shipped organoids maintained viability for at least 10 passages.

## Figures and Tables

**Figure 1 biomedicines-11-00151-f001:**
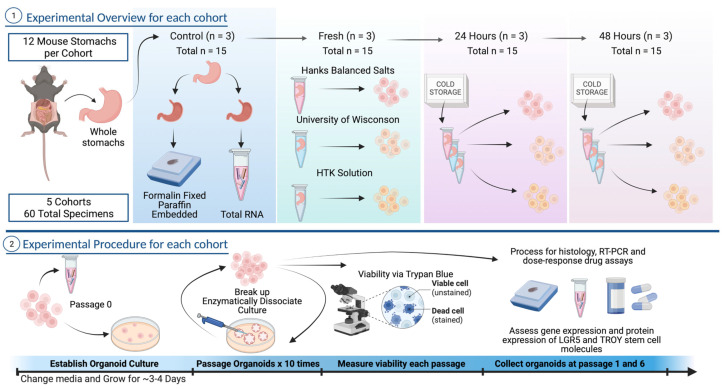
Study design. Images created with BioRender.com.

**Figure 2 biomedicines-11-00151-f002:**
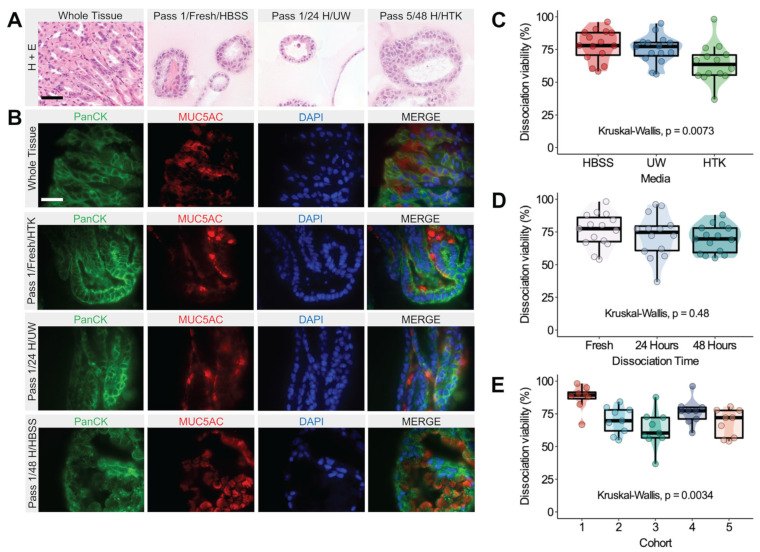
Gastric organoid histology and dissociation viability. (**A**) Representative hematoxylin and eosin images of whole mouse stomach and gastric organoids. The passage number, medium conditions, and dissociation times are noted in the image headings. Scale bar represents 50 µm. (**B**) Immunofluorescent images of whole mouse stomach and gastric organoids recapitulating epithelial pan cytokeratin (green) and gastric MUC5AC (red). Nuclei were stained with DAPI (blue). Images were captured using an AxioCam HRc camera and processed using ImageJ. Scale bar represents 20 µm. (**C**–**E**) Boxplot–scatter–violin plots representing univariable analysis for dissociation viability (percent of living cells) measured by Trypan Blue dye of all dissociated stomachs prior to organoid formation for medium (**C**), dissociation time (**D**), and cohort (**E**) conditions. Each scatter point represents one sample. The significance according to the Kruskal–Wallis test is presented in each respective plot.

**Figure 3 biomedicines-11-00151-f003:**
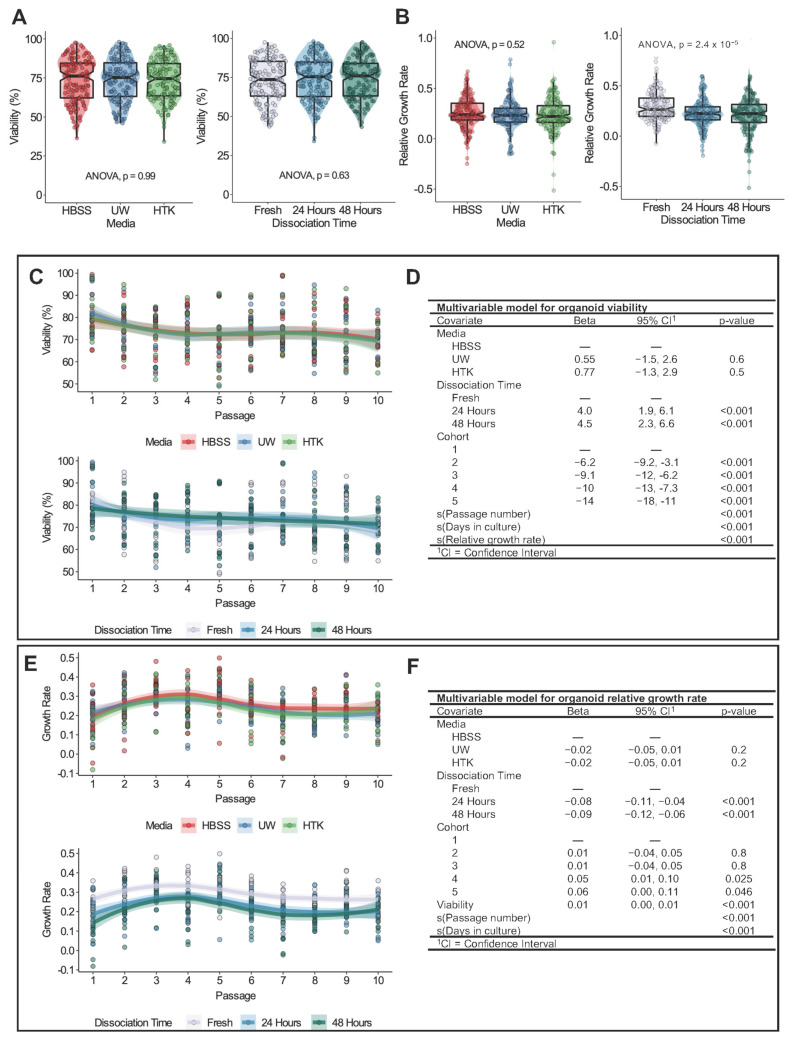
Gastric organoid viability and growth rate over 10 in vitro passages. (**A**) Boxplot–scatter–violin plots of percent viability versus medium (left) and dissociation time (right). (**B**) Boxplot–scatter–violin plots of relative growth rate versus medium (left) and dissociation time (right). For (**A**) and (**B**), the significance according to a one-way ANOVA test is presented in each plot. Colors in the plot legend indicate experimental conditions. Boxplot notches approximate the 95% confidence interval of the median. Each data point represents a single measurement. The distribution of data is represented by superimposed violin plots. (**C**) and (**E**) show adjusted response plots for viability (**C**) and growth rate (**E**) versus passage number stratified by medium and dissociation time. Adjusted viability estimates were predicted from the multivariable models (**D**,**F**). Each data point represents a single measurement. The trend in data was estimated using a loess smooth bound by semi-transparent 95% confidence intervals. (**D**,**F**) Regression table of generalized additive models for viability (**D**) and growth rate (**F**). Covariates are listed with accompanying beta coefficient values, 95% confidence intervals, and *p*-values. Covariates listed with s were fit using penalized cubic regression splines optimized by generalized cross-validation. Hyphens denote reference categories for discrete variables. Significant *p*-values are shown in bold.

**Figure 4 biomedicines-11-00151-f004:**
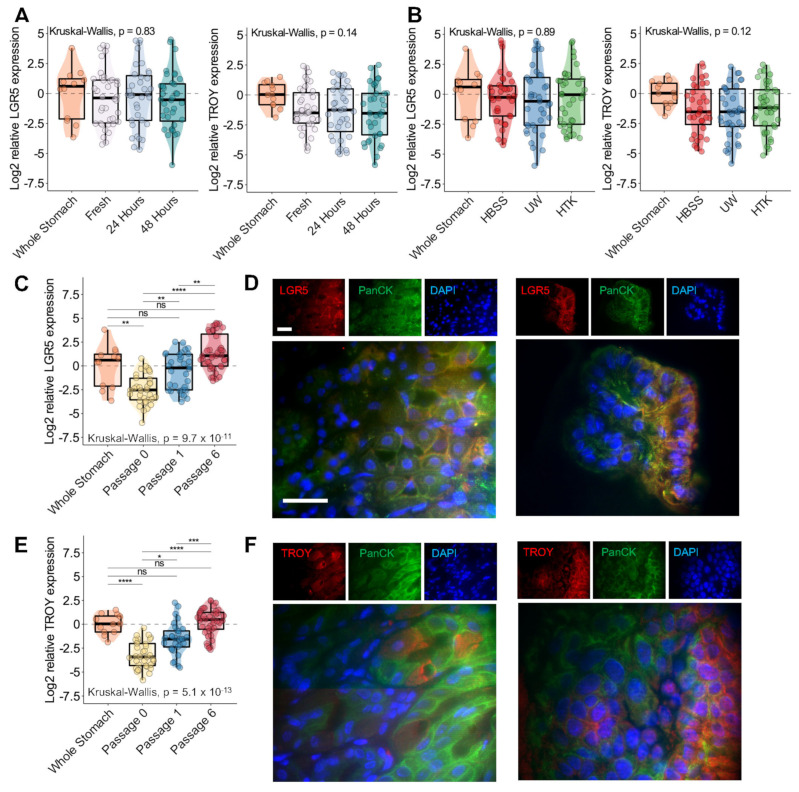
Expression of LGR5 and TROY stem cell markers in gastric organoids. (**A**) Boxplot–scatter–violin plots of log2 relative LGR5 gene expression (left) and log2 relative TROY gene expression (right) versus dissociation time. Whole stomach indicates control endogenous gene expression levels. (**B**) Boxplot–scatter–violin plots of log2 relative LGR5 gene expression (left) and log2 relative TROY gene expression (right) versus medium. (**C**) Boxplot–scatter–violin plots of log2 LGR5 gene expression versus passage number. The significance according to a Kruskal–Wallis and Dunn’s post hoc test is presented in the plot (**** *p* < 0.0001, *** *p* < 0.001, ** *p* < 0.01, * *p* < 0.05, ns = not significant). (**D**) Immunofluorescence expression of LGR5 (red) and pan cytokeratin (green) in whole mouse stomach and gastric organoids. Nuclei were stained with DAPI (blue). (**E**) Boxplot–scatter–violin plots of log2 TROY gene expression versus passage number. The significance according to a Kruskal–Wallis and Dunn’s post hoc test is presented in the plot. (**F**) Immunofluorescence expression of TROY (red) and pan cytokeratin (green) in whole mouse stomach and gastric organoids. Nuclei were stained with DAPI (blue). Images were captured using an AxioCam HRc camera and processed using ImageJ. Of note, for C and E, gene expression from control endogenous tissue and cells isolated from the initial dissociation are denoted by Whole Stomach and Passage 0, respectively. Gene expression for passages 1 and 6 was measured from established organoids. For D and F, the scale bar represents 20 µm.

**Figure 5 biomedicines-11-00151-f005:**
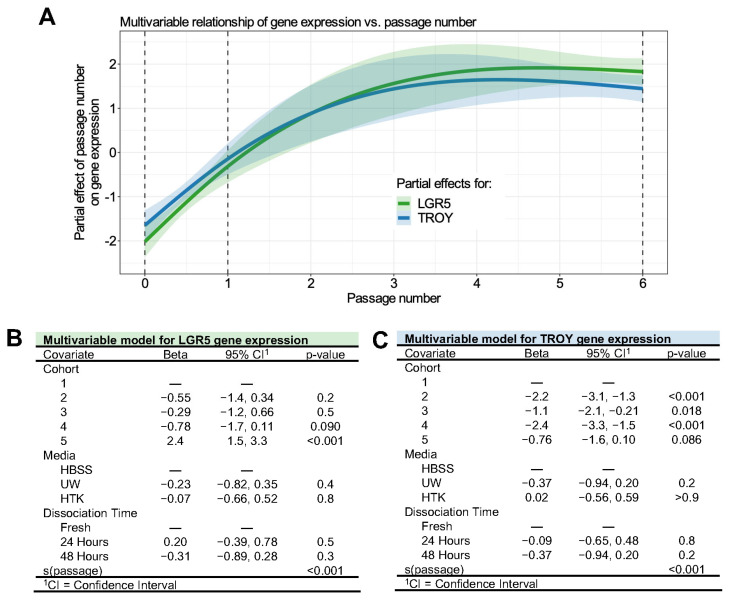
Multivariable analysis of LGR5 and TROY gene expression in gastric organoid culture. (**A**) Partial effects plot demonstrating the significant (*p* < 0.001) nonlinear effect of passage number on LGR5 (green) and TROY (blue) gene expression. The *y*-axis represents the change in gene expression for a given passage number as identified by the smoothed function. The solid colored lines are penalized cubic regression splines for passage numbers derived from multivariable models in B and C. Semi-transparent ribbons represent 95% confidence intervals. The dotted vertical lines represent the times at which gene expression was measured. (**B**,**C**) Regression tables of generalized additive models for LGR5 (C) and TROY (D) gene expression. Covariates are listed with accompanying beta coefficient values, 95% confidence intervals, and *p*-values where applicable. Covariates listed with s(passage) were fit using penalized cubic regression splines optimized by generalized cross-validation. Hyphens denote reference categories for discrete variables. Significant *p*-values are shown in bold.

**Figure 6 biomedicines-11-00151-f006:**
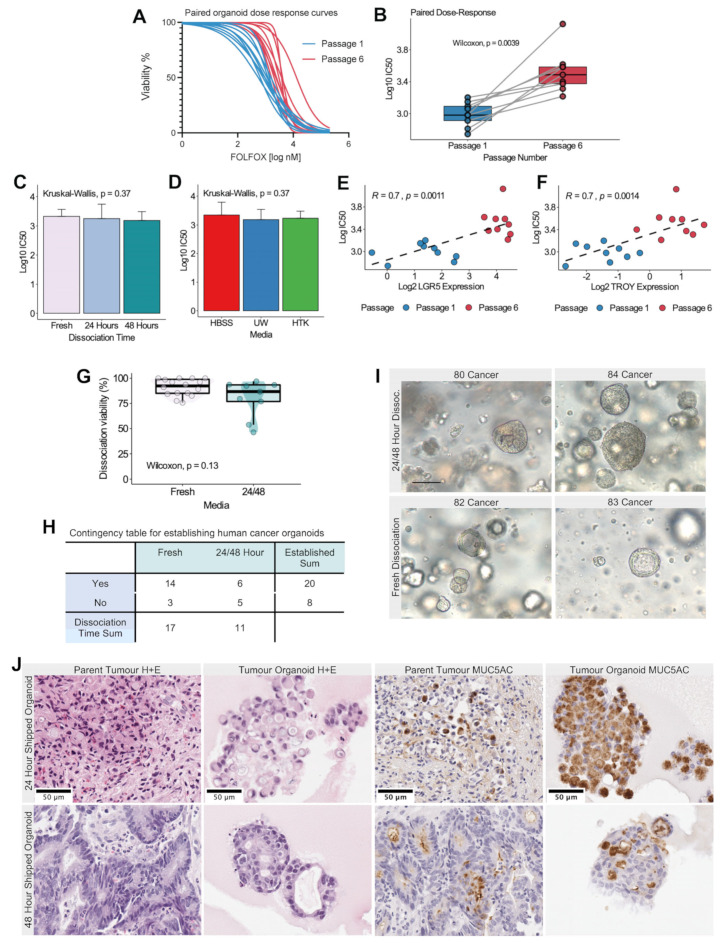
Effect of cold storage on dose response to FOLFOX therapy and growth of human gastric cancer organoids. (**A**) Dose–response curves for 9 organoid lineages in passage 1 (blue) and passage 6 (red). Cell viability measured by a CCK8 assay is presented on the *y*-axis and log10 FOLFOX concentration on the *x*-axis. Least-squares nonlinear models with variable slopes were fit for each of the 8 half-log dilution series. Half-maximal inhibitory concentration (IC50) was estimated from these models. (**B**) Paired boxplots for log10 IC50 values versus passage number. Each colored point and line correspond to a single organoid lineage. Organoids were derived for each set of experimental conditions (*n* = 9). Boxplots in grey represent the log10 IC50 distribution for all organoid lineages. Paired Wilcoxon *p*-value is included in the plot. (**C**,**D**) Barplots of Log10 IC50 values versus dissociation time (**C**) and media conditions (**D**). Experimental conditions are represented by colors identified in the plot legend. Kruskal–Wallis *p* values are presented in the plot. Error bars represent standard deviation. (**E**,**F**) Pearson’s correlation between log10 IC50 values and log2 relative gene expression for LGR5 (**E**) and TROY (**F**). Pearson’s R and *p*-values are presented in each plot. Points represent one organoid lineage and are colored according to passage number as indicated in plot legend below. The dotted black line represents the fit of simple linear regression. (**G**) Boxplot–scatter–violin plots for dissociation viability (percent of living cells according to Trypan Blue method) versus dissociation time for human gastric cancer specimens. (**H**) Contingency table for dissociation time versus successful organoid establishment for human gastric cancer specimens. (**I**) Gastric cancer organoids in vitro for cold-storage (top) and fresh (bottom) organoids, respectively. Image taken at 20x with AxioCam HRc. Scale bar represents 50 µm. (**J**) Representative brightfield images of human gastric cancer tumors and patient-derived organoids following 24 h (top) and 48 h (bottom) mock-shipment in cold HBSS. The image identification is presented in the plot heading. The first two columns show hematoxylin and eosin images. The third and fourth columns demonstrate the recapitulation of MUC5AC (brown) expression in tumor organoids. Images were captured using a Leica Aperio CS2 digital slide scanner and selected using QuPath version 0.3.1. Scale bar represents 50 µm.

**Table 1 biomedicines-11-00151-t001:** Multivariable linear regression for dissociation viability.

Covariate	Beta	95% CI ^1^	*p*-Value
Cohort			
1	—	—	
2	−11	−24, 1.2	0.075
3	−14	−32, 4.4	0.13
4	−10	−19, −0.72	0.036
5	−16	−26, −5.9	0.003
Age of Mouse	−0.18	−0.45, 0.10	0.2
Medium			
HBSS	—	—	
UW	−2.8	−9.8, 4.2	0.4
HTK	−14	−22, −6.8	<0.001
Dissociation Time			
Fresh	—	—	
24 Hours	−4.7	−12, 2.5	0.2
48 Hours	−6.8	−14, 0.18	0.056

^1^ CI = confidence interval.

## Data Availability

All raw data and code used to perform the analysis in this study are available publicly at https://github.com/skubleny/mouse-organoid 5 January 2022.

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
