# Peer review of "Murine and Human Gastric Tissue Establishes Organoids after 48 Hours of Cold Ischemia Time during Shipment"

_biomedicines, 2023, doi:10.3390/biomedicines11010151_

Round 1
Reviewer 1 Report
The article is devoted to a topical issue, obtaining organoids from tissue samples after long-term storage is of great interest both for scientific research and for personalized cancer therapy. The problem of delivering material from a hospital to a research laboratory is acute in all countries and can take up to 12 hours even within the same city. This study will attract the attention of many researchers. The work deserves publication, but the description of the design of the experiment needs to be corrected for greater clarity.
Minor:
Line 76-77: How this information related to experimental procedure?
Figure 1 is not clear. How many mice was in each cohort? What is the difference between cohorts? One stomach was divided or one stomach per condition was used? The corresponding text should also be made clearer.
"Organoid culture" : media composition is missed.
Author Response
Dear reviewer,
Thank you for your review and comments. Please see our following responses.
- Line 76-77: How this information related to experimental procedure?
Line 76-77 has been removed.
- Figure 1 is not clear. How many mice was in each cohort? What is the difference between cohorts? One stomach was divided or one stomach per condition was used? The corresponding text should also be made clearer.
Figure 1 and Study Design text have been updated for clarity
- "Organoid culture" : media composition is missed.
The supplement method has been incorporated into the main text to increase detail.
Reviewer 2 Report
3D tissue culture has become to a mainstay not only in cancer research but also in research to understand organ physiology and non-malignant diseases. Here, Skubleny and colleagues established a workflow to ship fresh gastric tissue to enable the preparation of gastric (tumor) organoids from tissue obtained by surgery in clinics not located nearby the cell culture lab. This study has the potential to facilitate the establishment of larger biobanks and the use of 3D tissue cultures in preclinical and clinical research.
The authors could show that storage for up to 48h had no influence on cell viability, drug sensitivity and expression of stem cell markers during cultivation. However, there were some inter-cohort differences and both stem cell expression and drug sensitivity changed significantly during passaging.
There are some issues to be resolved before publication.
General comments
The authors used mouse stomach as a model for various human gastric (tumor) tissues. For method establishment, homogeneous starting material is pivotal. However, the established method needs to be verified with real clinical samples and I wonder whether 2 patient derived tumors are sufficient to proof the applicability of the method on clinical samples.
Major comments:
Abstract: please include information about human samples (gastric tumor, n=2)
There was a significant difference in cell viability between different cohorts with effect sizes comparable to that of different storage solutions. This needs to be addressed.
Fig 2C: Control is missing. Please show viability after dissociation of stomachs directly after surgery without storage.
Fig 2 C-E: it is unclear which data is shown here. Does C comprise all 5 cohorts? All time points? Does D show all storage solutions combined? All cohorts? Etc.
Fig. 4B,C and E: what was the control?
Fig. 5A: Expression analysis was only done up to passage 6 although organoids were passaged at least 10 times. From the figure, one could speculate that expression of LGR1 and TROY reached a kind of plateau or steady state after passage 6. This could be clarified by showing all date up to passage 10.
Institutional Review Board Statement for animal experiments is missing.
Minor comments:
Line 21: delete repetition
Line 76: Please move this statement to a separate paragraph at the end of the manuscript.
Line 95ff: Please give the constitution of HBBS, UW and HTK (e.g. in the supplements)
Line 201f: Please revise this sentence
Author Response
Dear reviewer,
Thank you for your review and comments. Please see our following responses.
Re: General comments: Thank you for your comment. We have added additional clinical specimens which we have grown using this cold storage method. In total there are now 5 human gastric cancer organoids. 3 of the organoids are currently in culture and we have included in-vitro images of these organoids.
- Abstract: please include information about human samples (gastric tumor, n=2)
The number of human subjects has been updated.
- There was a significant difference in cell viability between different cohorts with effect sizes comparable to that of different storage solutions. This needs to be addressed.
We have added another paragraph in the discussion to address this interesting finding.
- Fig 2C: Control is missing. Please show viability after dissociation of stomachs directly after surgery without storage.
Re Fig 2C: All stomach tissues, even those processed fresh, were placed in storage media conditions. We have modified Fig 2C,D,E to represent all organoids that were dissociated with enough cells to quantify. To provide additional clarity on these effects we add Figure S3 and S4 which show the dissociation viability of the Fresh samples only. We also show the breakdown of dissociation viability for all three time points with respect to the media condition.
- Fig 2 C-E: it is unclear which data is shown here. Does C comprise all 5 cohorts? All time points? Does D show all storage solutions combined? All cohorts? Etc.
We have updated the figure legend and modified the figure to show each individual data point. These show all dissociated organoids.
- 4B,C and E: what was the control?
For Figure 4A,C and E the control is the undissociated stomach. The figure legend states that “Whole stomach indicates control endogenous gene expression levels”. In Figure 4B we have added the undissociated control into the plot for consistency.
- 5A: Expression analysis was only done up to passage 6 although organoids were passaged at least 10 times. From the figure, one could speculate that expression of LGR1 and TROY reached a kind of plateau or steady state after passage 6. This could be clarified by showing all date up to passage 10.
Although we are able to satisfy all other reviewer comments, we cannot provide data for passage 10 because samples were not obtained for this time point. Certainly, this is an interesting question that should be pursued in future research. Do stem cell markers plateau or do they progressively increase? At what rate? Is it dose dependent on L-WRN concentration?
- Institutional Review Board Statement for animal experiments is missing
Added to the appropriate location.
Reviewer 3 Report
In this manuscript, the authors investigated the influences of various shipping media or dissociation times on the generation of normal gastric organoids in mice and gastric cancer organoids in humans. They found that the organoids could be established after 24 and 48 hours of storage with ice packs. Standard HBSS solution is feasible and reliable. This method does not affect the reliability of downstream dose-response assays and maintains organoid viability. The manuscript is well-arranged, and the logic is clear. So I recommend this manuscript could be accepted after minor revision.
The following comments need to be addressed:
1. Why can intestinal organoids be established through murine and human gastric tissue? Is it a mistake? Gastric organoids are not equivalent to intestinal organoids.
2. There are few human gastric tissue samples for the study. If authors could provide more samples from different origins?
3. Dissociation viability is affected by storage media and is cohort dependent. How about organoid formation efficiency?
Author Response
Dear reviewer,
Thank you for your review and comments. Please see our following responses.
- Why can intestinal organoids be established through murine and human gastric tissue? Is it a mistake? Gastric organoids are not equivalent to intestinal organoids.
We will clarify the document. They are all gastric organoids. All references to intestinal organoids have been removed.
- There are few human gastric tissue samples for the study. If authors could provide more samples from different origins?
We have added additional clinical specimens which we have grown using this cold storage method. In total there are now 6 human gastric cancer organoids developed following cold storage. We have included in-vitro images of these organoids and dissociation viability data.
- Dissociation viability is affected by storage media and is cohort dependent. How about organoid formation efficiency?
Correct. We state that all experimental conditions successfully established organoid culture. We do not have specific time data for exactly when a cystic organoid structure was created. However, we feel that if the organoid is able to be established, passaged and utilized in experimental contexts without consequence (as demonstrated by our experiments) it follows that any differences in the actual time to form an organoid is not practically significant. Of course, additional research could prove otherwise, as functional hormonal performance of the organoid was not evaluated in this study.
Round 2
Reviewer 2 Report
Thank you for your revisions!
I wish you a happy an dprosperous 2023.